# Deciphering Tumour Microenvironment of Liver Cancer through Deconvolution of Bulk RNA-Seq Data with Single-Cell Atlas

**DOI:** 10.3390/cancers15010153

**Published:** 2022-12-27

**Authors:** Shaoshi Zhang, Wendi Bacon, Maikel P. Peppelenbosch, Folkert van Kemenade, Andrew Peter Stubbs

**Affiliations:** 1Department of Pathology and Clinical Bioinformatics, Erasmus University Medical Centre, Wytemaweg 80, 3015 CN Rotterdam, The Netherlands; 2Department of Gastroenterology and Hepatology, Erasmus University Medical Centre, Wytemaweg 80, 3015 CN Rotterdam, The Netherlands; 3School of Life, Health & Chemical Sciences, Faculty of Science, Technology, Engineering & Mathematics, The Open University, Milton Keynes MK7 6AA, UK

**Keywords:** liver cancer, tumour microenvironment, deconvolution

## Abstract

**Simple Summary:**

ScRNA-seq is a powerful tool for investigating the cancer microenvironment, but the cost of analysing every scientific scenario is prohibitive. Fortunately, deconvolution of bulk RNA-seq data with scRNA-seq cell atlas reference datasets provides a cheaper strategy. In this study, we validated the feasibility of deciphering the microenvironment of liver cancer through the estimation of cell fractions with Cibersortx and scRNA-seq atlas reference datasets. Five cell types are associated with patient outcomes, showing that deconvolution is a useful method for characterising the tumour microenvironment.

**Abstract:**

Liver cancers give rise to a heavy burden on healthcare worldwide. Understanding the tumour microenvironment (TME) underpins the development of precision therapy. Single-cell RNA sequencing (scRNA-seq) technology has generated high-quality cell atlases of the TME, but its wider application faces enormous costs for various clinical circumstances. Fortunately, a variety of deconvolution algorithms can instead repurpose bulk RNA-seq data, alleviating the need for generating scRNA-seq datasets. In this study, we reviewed major public omics databases for relevance in this study and utilised eight RNA-seqs and one microarray dataset from clinical studies. To decipher the TME of liver cancer, we estimated the fractions of liver cell components by deconvoluting the samples with Cibersortx using three reference scRNA-seq atlases. We also confirmed that Cibersortx can accurately deconvolute cell types/subtypes of interest. Compared with non-tumorous liver, liver cancers showed multiple decreased cell types forming normal liver microarchitecture, as well as elevated cell types involved in fibrogenesis, abnormal angiogenesis, and disturbed immune responses. Survival analysis shows that the fractions of five cell types/subtypes significantly correlated with patient outcomes, indicating potential therapeutic targets. Therefore, deconvolution of bulk RNA-seq data with scRNA-seq atlas references can be a useful tool to help understand the TME.

## 1. Introduction

Liver cancer is one of the leading causes of cancer-related mortality worldwide, making up 4.7% of newly diagnosed cases but 8.2% of deaths [1]. Hepatocellular carcinoma (HCC) and cholangiocarcinoma (CCA), which are frequently tallied together, constitute the major burden of liver cancer [2]. With a 5-year survival of 18%, liver cancer ranks as the second-most lethal cancer [3]. The poor prognosis of liver cancer is partially due to the insufficiency of effective therapies. Surgical interventions yield the best outcomes but are limited by high recurrence rates or easy loss of operative windows. Liver transplantation achieves better long-term survival but is hampered by an inadequate supply of donor organs [4,5]. Systematic non-specific chemotherapy resulted in disappointing results for both HCC and CCA [5,6]. Innovative agents targeting angiogenesis [7,8], fibrogenesis [9], and regulation of immune responses [10,11] have shown the potential to improve outcomes. Such advances suggest that a shift from a cancer-centric to a tumour microenvironment (TME)-centric perspective is required in the future development of precision therapy [12,13].

Single-cell sequencing (scRNA-seq) technology delivers in-depth interrogation of the TME. The analysis of complex cancer tissues at the single-cell level through scRNA-seq has brought insights into the heterogeneity and progression of cancer, as well as escape from immune surveillance, drug resistance, and intercellular communication [14]. However, scRNA-seq technology is expensive, requires specific tissue collection methods, and can be difficult to implement. The cheaper and more common bulk RNA-seq studies make up the largest body of work in this area, filling public repositories, including flagship projects such as The Cancer Genome Atlas (TCGA) and its resulting resource the Pan-Cancer Atlas [15]. To make the most out of these available datasets, numerous algorithms have been proposed that enhance the informativeness of bulk transcriptome analysis using reference scRNA-seq data. Machine learning is the major approach of such methodologies. For example, stemness indices within the Pan-Cancer Atlas were estimated with the Progenitor Cell Biology Consortium datasets and one-class logistic regression [16]. Support vector regression was also used to estimate the abundance of a specific component in bulk RNA-seq samples [17,18]. Another group of deconvolution algorithms, such as MuSiC [19] and Cibersortx [20], focus on profiling cell fractions of the bulk transcriptomic data. Meanwhile, emerging scRNA-seq atlases (e.g., Human Cell Atlas) provide high-quality references, allowing accurate deconvolution of bulk RNA-seq data in increasingly wider contexts (e.g., profiling of TME for head and neck cancers [21]).

In this study, we reviewed major omics databases and selected ten studies that compared transcriptomes between HCC/CCA and normal liver. The cell fractions of tumour and non-tumour tissues were estimated with Cibersortx and three scRNA-seq reference atlases. The included studies contain two cohort studies of HCC, allowing us to investigate the clinical implications of TME abnormalities through survival analysis and gene set enrichment analysis (GSEA) [22]. We determined that the TMEs of liver cancer lack multiple cell types (e.g., hepatocytes) that form normal liver microarchitectures, and instead are enriched with components involved in fibrogenesis, abnormal angiogenesis, and irregular immune activities. Among all the cell types/subtypes in the HCC TME, hepatocytes and mature B cells are positively correlated with patient outcomes, while cholangiocytes, bi-potent stem cells, plasma B cells, and regulatory T (T_reg_) cells correlate with negative outcomes.

## 2. Materials and Methods

### 2.1. Data Obtainment

We searched three public omics databases—the Gene Expression Omnibus (GEO), The Cancer Genome Atlas (TCGA), and the International Cancer Genome Consortium (ICGC)—for studies engaging liver cancers. Inclusion criteria were: (1) histologically identified sample series from human tissues in clinical studies (including cohort studies and case review series); (2) whole transcriptomes by microarray or RNA-seq; (3) with non-tumorous or normal liver tissues as controls (for comparison) or follow-up information (for interrogation of clinical outcomes); (4) studies covering two major liver cancers (HCC and CCA) were included and all other studies were excluded, i.e., metastatic liver cancers and non-malignant hyperplasia (e.g., hemangioma). Finally, nine curated datasets (eight RNA-seq and one microarray) were selected for this study (Appendix A).

### 2.2. Pre-Processing of Microarray Data

Microarray studies with raw data (CEL files) from the GEO database were obtained via R package SCAN.UPC [23], which provides one-step pre-processing through empirical correction of major bias (GC content-related). This normalisation method for microarray datasets proves reliable for downstream analysis [24]. All gene names were translated into HGNC symbol with R package BioMart [25]. Duplication of features in expression matrices were collapsed by MaxMean strategy [26].

### 2.3. Pre-Processing of RNA-Seq Data

TCGA-LIHC was obtained via UCSCXenaTools [27] and datasets in ICGC were obtained via the official web portal [28]. Expression matrices of other studies from the GEO database were retrieved according to the authors’ instructions. All datasets recruited in this study have been listed in Appendix A. All gene names were translated to HGNC symbols with R package BioMart [25]. Feature duplications in expression matrices were removed with the summation method [26].

### 2.4. Deconvolution of Cell Types with Cibersortx and Three Atlases

Three single-cell RNA-seq datasets were used in this study: (1) GSE115469 (Normal Liver), (2) GSE146409 (TME-Stroma), and (3) GSE156337 (TME-Immune). GSE115469 is a liver subset of the Human Cell Atlas [28]. GSE146409 contains the TME of liver cancer (HCC and CCA), including malignant cells [29]. GSE156337 contains the HCC microenvironment, without malignant cells [30]. This dataset identified high-quality immune cells.

All expression matrices were normalised to 10,000 counts/cell and packed into an H5AD file with authors’ annotation as metadata for subsequent estimations. Both the signature matrix of scRNA-seq datasets and the expression matrix of bulk RNA-seq datasets were transformed into tab-delimited tables. The signature matrices of reference scRNA-seq were built with the Cibersortx protocol for “scRNA-seq” (“S”). Deconvolution was performed with the “Impute Cell Fractions” module. In validation experiments, batch correction was disabled in within-study validation and activated in “S” mode (with single-cell expression matrix as reference) in cross-study validation (two groups of validation experiments will be described below). In the estimation of real clinical data, batch correction was enabled in “S” mode for RNA-seq datasets and “B” mode (with single-cell expression matrix collapsed into bulk matrix before used as reference) for microarray datasets. All other Cibersortx parameters were the default configurations [20].

Cibersortx fails when the cell type tree of the reference atlas is complicated (e.g., >10 cell types). In this situation, collapsing some branches of the cell type tree would complete the calculation [20]. In our study, for example, when we used the Normal Liver atlas, the cell type tree was divided into two groups (immune and non-immune groups). Similarly, the TME-Stroma atlas was divided into three groups (mesenchymal, vasculature and immune groups), and the TME-Immune atlas was divided into two groups (immune and non-immune groups) (All cell type trees are shown in Appendix A). To evaluate the influence of adjusting cell type tree, we performed a group of validation experiments in which cell subtypes in the Normal Liver atlas were collapsed (for example, let macrophage = inflammatory macrophage + non-inflammatory macrophage, T cell = alpha-beta T cell + gamma-delta T cell, B cell = mature B cell + plasma B cell, and LSEC = periportal LSEC + central venous LSEC). To increase the matching in cross-study experiments (described below), the cell-type trees of the other two atlases were also adjusted (for TME-Stroma atlas, let macrophage = scar-associated macrophage + Kupffer cell + tissue macrophage, and for TME-Immune atlas, let T cell = CD4^+^ T cell + CD8^+^ T cell + T_reg_ cell).

### 2.5. Accuracy and Robustness of Cell Fraction Estimation

The accuracy and robustness of deconvolution with Cibersortx were tested in two groups of experiments: intra-study validation and cross-study validation. In intra-study validation experiments, the signature matrix of scRNA-seq and the pseudo-bulk for testing were generated from the same scRNA-seq atlas. The advantage of this method is full coverage of all cell types/subtypes. In cross-study validation experiments, the signature matrix of scRNA-seq and the pseudo-bulk dataset were generated from different atlases.

Pseudo-bulk datasets were generated using the random module of NumPy in the following procedure: the expression of the scRNA-seq atlas was separated into two groups, cell type of interest and all the remaining cell type/subtypes collapsed into a single group labelled as “others”; 10% of cells in each group were selected using Choice function of NumPy, then two representative expression vectors (V) were generated by calculating the mean value of each gene; a random number f (between 0 and 100) was generated by Uniform function of NumPy; finally, the expression vector of the pseudo-bulk sample was generated by $V_{celltype}\times f+V_{others}\times \left(100-f\right)$. f was used as benchmarking target.

Two parameters were used to evaluate the accuracy of the Cibersortx estimation: (1) Pearson correlation coefficient (PCC) between predefined proportions (f) and estimated fractions ($f^{\prime }$); (2) mean absolute error (MAE = $1/n\sum_{i=1}^{n}\left|f^{\prime }{}_{i}-f_{i}\right|,i=1,...,n$) and direction (overestimation or underestimation).

### 2.6. Survival analysis, Statistics, and Data Visualisation

To demonstrate the added value of our deconvolution analysis, we investigated the survival impact of estimated cell fractions on two HCC cohorts (TCGA-LIHC, GSE14520). The patient cohort was first ordered based on descending order of estimated cell fractions and then separated into high- and low-level groups. All separation possibilities (from 1:n-1 to n-1:1) were tested with log-rank tests. The one with the lowest P-value in log-rank tests was selected as the optimised separation. If all the P-values were above 0.05, the cohort was equally separated into two groups (median-point separation).

Survival lengths were transformed into months and observed events (labelled as “1”) were transformed into “True” (Boolean value, according to the requirement of Scikit-Survival [31]). Kaplan–Meier (K-M) curves were then fitted with Scikit-Survival [31] and plotted with the Step function of Matplotlib. The log-rank test was calculated with Lifelines [32]. All boxplots of this study were generated with MatPlotLib.

Pathway activities were estimated by PROGENy [33]. GSEA was performed with GSEApy. The input gene list for GSEA was the marker genes selected by Cibersortx for the cell types. Our study shows a demonstration with “WikiPathway 2021—Human” as the reference. GSEApy allows dozens of different libraries, and we provide scripts in our GitHub repository.

For better reproducibility, all the analysis scripts including pre-processing have been shared with GitHub (https://github.com/ErasmusMC-Bioinformatics/OIO_Shaoshi, accessed date 27 December 2022). The supplements and pre-processed datasets were shared with Zenodo (https://doi.org/10.5281/zenodo.7467268, accessed date 27 December 2022).

## 3. Results

### 3.1. Cibersortx Estimation of Cell Fraction

In this study, we aim to decipher the TME of liver cancer by estimating the cell fractions in transcriptomes. This requires an accurate and robust model with well-annotated single-cell atlases. We adopted a state-of-the-art deconvolution algorithm (Cibersortx) and three scRNA-seq atlases (Normal Liver, TME-Stroma, and TME-Immune). These atlases describe more than 20 cell types or subtypes. Figure 1A outlines the workflow of this study, and the cell-type trees of the three atlases are outlined in Appendix A.

We first performed intra-study validation experiments to test whether all identified cell types from each atlas could be accurately deconvoluted with Cibersortx. In this group of experiments, pseudo-bulk datasets were generated by the same atlas (Figure 1). In this validation mode, most cells achieved ideal PCCs (Appendix A). The PCCs compare the relationship between predefined cell fractions in generated pseudo-bulk samples and estimated values by Cibersortx. However, MAEs vary between different cell types (Appendix A). To evaluate the effect of merging some cell types, we did a group of tests by combining the major subtypes in the Normal Liver atlas (Appendix A). Both panels in Figure 1B show the result of intra-study validation with the Normal Liver atlas. Although both experiments show high levels of PCCs, subtle differences in accuracy exist. For example, after merging eight cell subtypes into four major cell types, the PCC of hepatocyte increases (0.979 vs. 0.992, Appendix A, the same the following), cholangiocyte decreases (0.9915 vs. 0.9844), and hepatic stellate cell (HSC) decreases (0.9901 vs. 0.986).

Although Cibersortx allows “partial deconvolution”, in which the samples may contain cell type/subtypes not present in the reference atlas, we performed a group of cross-study validation experiments to assess this impact. The results of these experiments demonstrate that the PCC between the presets and prediction remain high (Figure 1E, Appendix A, Page 19) whilst the MAEs vary significantly between cell types (Appendix A, Page 19). Our study suggests that Cibersortx normally guarantees high levels of PCCs but MAEs vary when using partial deconvolution method. Thus, we adopted a protocol in all subsequent analyses such that if a cell type (e.g., hepatocytes) was present in multiple atlases, the one with the best performances in both intra-study and cross-study validation experiments was chosen as the reference for clinical data.

### 3.2. Difference of Cell Fraction between Tumour and Non-Tumour Liver Tissue

To determine the difference in cell fraction between tumour and non-tumour tissues, we compared seven RNA-seq datasets which provide paired tissues collected from HCC/CCA cohort studies or case review studies. LIRI-JP is an RNA-seq study that includes primary liver cancers, and secondary liver cancers from stomach, colon, prostate, etc., with adjacent non-tumorous liver tissues as controls. GSE119336 is an RNA-seq study comparing CCA and non-tumour liver. The other five RNA-seq studies compare HCC and non-tumour liver tissues. Three of these studies included cases with HBV infection as the risk factor.

The three atlases provide more than 20 cell types, allowing a panoramic view of TMEs. All of these cell types can be largely divided into three groups: (1) cell types underpinning the fundamental physiology of livers, e.g., hepatocytes, cholangiocytes; (2) cell types participating in the pathological evolution of cancer formation, e.g., HSCs, LVECts; (3) immune cells. Results of related cell types are described in the following groups.

#### 3.2.1. Hepatocytes and Cholangiocytes

These two cell types constitute the major functional units of livers—liver lobule and bile ducts [28]. As both the Normal Liver and TME-Immune atlases have hepatocytes, we made a comparison of results by deconvolution with two different atlases for the same dataset. Figure 2A,B show the results of deconvolution for the same datasets with different atlases. Compared with the results determined with the TME-immune atlas, deconvolution with the normal atlas returned higher levels of hepatocytes for some datasets (such as GSE119336) and lower levels for others (such as GSE94660). However, both experiments arrived at the same result whereby the decrease of hepatocytes and elevation of cholangiocytes are common in liver cancer. Similarly, Appendix A, pages 1–3 generated by deconvolution with TME-Stroma resulted in the same conclusion. Compared with HCC, the decrease of hepatocytes in CCA is more significant (Figure 2A,B).

#### 3.2.2. Fibrogenesis

HSCs are tissue-specific equivalents of pericytes, and pericytes can be rarely detected in normal livers [29,34]. Fibroblasts proliferate following the activation of HSCs. Compared with non-tumorous tissues, a decrease in HSCs and the elevation of pericytes can be seen in liver cancers. CAFs are rarely detected in normal liver and are more often seen in liver cancers. The opposite alterations between HSCs and pericytes/CAFs suggest active fibrogenesis in tumours (Figure 2D–F).

#### 3.2.3. Vasculature

Liver sinusoidal endothelial cells (LSECs) form the wall of hepatic sinusoids and participate in immune responses. Vascular smooth muscle cells (vSMCs) are also key components of blood vessels. Tumour liver vascular endothelial cells (LVECts) are variants of normal LVECs [29]. Decreased LSECs, vSMCs, and LVECs can be seen in cancerous tissues, while the elevation of LVECt can be observed in liver cancers (Figure 3A–D). These results suggest the emergence of abnormal angiogenesis.

#### 3.2.4. Immune Cells

T cells: In the three atlases, five subsets of T cells were identified: T cells with alpha–beta (αβ) TCR chains or gamma–delta (γδ) chains (Normal Liver atlas) [28]; CD4^+^ (helper), CD8^+^ (cytotoxic) T cells, and T_reg_ cells (TME-Immune atlas) [30]. We observed obvious elevations of αβ T cells but no clear alterations of γδ T cells in liver cancers. CD4^+^ T cells rise sharply in CCA while CD8^+^ T cells elevate moderately in HCC. Finally, T_reg_ cells are rarely detected in normal livers whereas elevations are common in liver cancers. T_reg_ cells have been recognised as a suppressor of tumour immune responses. Liver cancers also show high levels of overall T cells (CD4^+^+CD8^+^+T_reg_ cells). (Figure 4A–F).

B cells: The Normal Liver atlas identified two subtypes of B cells, mature B cells (antigen inexperienced) and plasma B cells (antigen secreting) [28]. Variation of mature B cells do not show a direct association with liver cancers. Plasma B cells were rarely detected in normal liver but were widely detected in liver cancers (Figure 5A,B).

Macrophages: The Normal Liver atlas contained inflammatory and non-inflammatory macrophages [28]. The TME-Stroma atlas separated Kupffer cells, tissue monocytes (TMs), and scar-associated macrophages (SAMs) [29]. SAMs are often recruited in the process of liver fibrosis [29,35]. We observed elevations of inflammatory macrophages and SAMs in liver cancers. No obvious differences in Kupffer cells, TMs and non-inflammatory macrophages were seen (Figure 5C–F).

Dendritic cells: As the quintessential antigen-presenting cell, the dendritic cell (DC) is another hot research interest for the potential of immunotherapy. The TME-Stroma atlas was used to estimate conventional DC1 and DC2 (cDC1 and cDC2) cell types [29,36]. Deconvolution suggests that cDC1 cells are elevated in liver cancers while cDC2 cells are not (Figure 6A,B).

In summary, liver cancers show higher levels of overall immune cells, involving both the innate (monocyte–macrophages) and adaptive branches (T, B cells), as well as auxiliary components (dendritic cells). Meanwhile, suppressive components such as T_reg_ cells can also be observed, suggesting the disordered responses in tumours.

#### 3.2.5. Bi-Potent Stem Cells and Proliferative Cells

This group involves two cell types that can proliferate and differentiate. Bi-potent stem cells (from TME-Immune atlas) were named for their potential to differentiate into both hepatocytes and cholangiocytes [30]. Proliferating cells were identified in the TME-Stroma atlas and elevation of these two cell types (HCC and CCA) was common in tumours [28]. Bi-potent cells were rare in normal livers and their elevation in CCA is prominent (Figure 6C,D).

#### 3.2.6. Other Cell Types

These three atlases also identified some immune cell types which exist in the liver with a relatively low abundance. TME-Immune identified a cluster of natural killer (NK) cells (a key component of the innate immune branch) and a cluster of myeloid cells (the liver-resident precursors of monocytes–macrophages) [30]. The Normal Liver atlas isolated a cluster of NK-like cells, which may be an ambiguous mixture of natural killer T (NKT) cells and NK cells [28]. Different atlases may have some cell types/subtypes with the same labels. However, calculated signature matrices suggest that they have different scopes, e.g., HSCs in the Normal Liver atlas, Figure 2D, vs. HSCs in the TME-Stroma atlas, Appendix A, Page 1).

### 3.3. Cell Fraction of HCC TME Correlates with Clinical Outcome

Finally, we investigated whether cellular alteration affects clinical outcome of HCC through survival analysis. In public repositories, TCGA-LIHC is the highest-cited cohort of a liver cancer study, with well-annotated follow-up information and substantial sample size (370 HCC patients). TCGA-LIHC is a pooled study of five cohorts with mixed risk factors. Available survival analyses include overall survival (OS) and disease-free survival (DFS) [37].

The distributions of estimated cell fractions show two typical shapes, “Sigmoid” or “Exponential” (Appendix A). Using an optimisation strategy (lowest log-rank test p-value), the patient cohort was typically separated at inflection points, although this separation may fail in cases of negative results or meaningless grouping (e.g., separating one case into a group). In these circumstances, we used the median-point strategy to finish complete KM curves (Figure 7, Appendix A).

Among all the estimated cell types, hepatocytes and bi-potent cells show substantial impacts on patients’ outcomes. The estimated fractions of hepatocytes show a sigmoid-shaped distribution. The optimisation strategy (lowest log-rank test p-value) separates the cohort at a close-to-median point in OS analysis, and at an inflection point for DFS analysis (Figure 7C). High fractions of hepatocytes are associated with longer OS and DFS (Figure 7A,B). Estimated fractions of bi-potent cells show exponential distribution. The optimisation strategy isolated a subset with the cell fractions close to zero (Figure 7F). Those with high fractions of bi-potent cells show lower OS and DFS (Figure 7D,E). TCGA also included serum alpha–fetoprotein (AFP) at patient admission. Although the differences of OS and DFS between patients with high- or low-level AFP reach statistical significance, crossings of KM curves exist. In contrast, deconvoluted hepatocytes and bi-potent cells show better discrimination (Figure 7).

GSE14520 is a study that recruited more than 200 HBV-related HCC cases and provides both OS and DFS information [38,39]. Its transcriptomic tests are based on microarray platforms, which may provide less accuracy than RNA-seq. In our study, rare cell types were not often detected in the deconvolution of microarray data. However, given the subtle difference of study protocol, it still provided alternative evidence about the impact of cell fractions on patients’ outcomes. A compilation of the survival analyses for GSE14520 is available in Appendix A, with a summary in Appendix A. Consistent and significant results for both OS and DFS in the two cohorts include: hepatocyte (positive), cholangiocyte (negative), bi-potent stem cell (negative), mature B cell (positive), plasma B cell (negative), and T_reg_ cell (negative).

Pathway analysis provides useful information for the identification of therapeutic targets. Figure 8 shows the estimation of pathway activities by PROGENy [33]. Only a small cell population show high activities of specific pathways. For example, LVECt, proliferating cells, bi-potent cells, and mast cells are EGFR active. GSEA is another useful method. Appendix A shows the examples using the signatures from the three atlases generated by Cibersortx with the library “WikiPathway 2021—Human” as the source of pathway definitions.

### 3.4. Cell Abundance Estimation by Support Vector Regression

We also conducted a parallel series of analyses by support vector regression. Appendix A describes the detailed methodology. Both methods achieved the same conclusions for most cell types. Of note, a subtle difference exists between the prediction by SVR and Cibersortx deconvolution. SVR estimates abundance between samples while the summation of deconvoluted fractions is equal to one. A substantial alteration of mass cell types (most often hepatocytes in liver) may significantly impact trace components in deconvolution.

All Signature matrices and scripts have been shared online (see GitHub/Zenodo address).

## 4. Discussion

Deconvolution algorithms draw high interest in estimating cell fractions with scRNA-seq atlases. Cibersortx and MuSiC are two state-of-the-art algorithms in benchmarking studies [40,41]. We adopted Cibersortx for better reproducibility while implementation with MuSiC involves R scripting and manual selection of markers. In our study, in silico experiments determined that Cibersortx can accurately predict the fractions of cell types. We also performed a parallel series of analyses with SVR to reassure the conclusions by Cibersortx.

Cibersortx estimates cell fractions by quantifying the abundance of signature genes, which warrants careful consideration when interpreting the results. Cell fractions can be defined in diverse ways. In conventional histological studies, cell proportion calculates by volume, cell number, mass, etc., [42]. The prediction of Cibersortx seems to be close to the definition of “fraction by cell number”. However, the expression of signature genes varies between cells, tissues, individuals, and the different disease states, leading to an ambiguity in the concept of “fraction”. This gap becomes prominent when the biomedical conditions of the reference scRNA-seq atlases and the bulk RNA-seq samples differ (necessitating “partial deconvolution”). In addition, cell clusters between atlases with the same label may not be identical. In our study, we preserved all the signature matrices generated by Cibersortx for better comparison. Therefore, we recommend taking into account all of these factors when interpreting the biomedical implications of the deconvoluted results [20].

Three scRNA-seq atlases help portray biological events in liver cancers, including angiogenesis, fibrogenesis, immunity, and stem cell transformation. Agents targeting angiogenesis made the first breakthrough in liver cancer chemotherapy [43]. Cancer growth signifies abnormal blood supply in the TME, with simultaneous pseudo-hypoxia and neo-vasculature. The TME-Stroma atlas demonstrated the involvement of pericytes and LVECt in tumour angiogenesis [29]. As vessel components and immune barriers, LSECs lose their position in tumour tissues. The retreat of LSECs gives way to metastasis [44]. Our study shows that these alterations are common in liver cancers.

Liver carcinogenesis has close ties to chronic inflammations, either viral hepatitis or steatohepatitis. A normal liver resides liver-specific pericytes (also termed hepatic stellate cells), which can be inflammation-activated. Protracted inflammation leads to stellate cell transformation and fibroblast proliferation [45]. The latter interacts with malignant cells and helps a tumour-favourable microenvironment. Cancer-associated fibroblasts (CAFs) provide an immune-evading and chemotherapy-resistant barrier. They also secret multiple cytokines promoting tumour growth and angiogenesis [46]. Experimental evidence shows that CAF remodelling of extracellular matrix (ECM) helps the metastasis-promoting TME [47,48]. Thus, therapeutic strategies targeting CAFs are in development, such as nanocarriers [9]. Our study shows the universal attendance of CAFs in liver cancers, indicating the potential of CAF-targeting therapies.

Liver cancer evolves with the cross-talk between malignant cells and the immune system. In the early phase of cancer initiation, immune cells actively move towards and fight against transformed cells. Long-term engagement finally resulted in the exhaustion of anti-tumour immunity. Some of the immune cells serve the malignant transformation. Such components include well-established tumour-associated macrophages [49,50], exhaustive and immunosuppressive T cells [51,52], and gradually recognised tumour-infiltrating B cells [53]. Our study provides a helicopter view of the broad attendance of immune cells in tumour tissues and the appearance of unfavourable components such as T_reg_ cells. T_reg_ cells function through the PD-1/PD-L1 pathway, leading to tumour tolerance. A blockade of this communication results in the resurrection of immune responses in a minor group of patients. Further studies show that the therapeutic efficiency of a PD1 inhibitor depends on the interaction between the TME and other immune components (e.g., CD8^+^ T cells) [10,54].

The cancer stem cell (CSC) hypothesis proposed that a small cell population harbouring embryonic characteristics fuel tumour growth. It is difficult to identify CSCs except by tracing their descendants bearing specific features. Reported biomarkers of liver CSCs include CD133, CD90, epithelial cell adhesion molecule (EpCAM), etc., [55]. EpCAM^+^ cells were proposed as a tumour-initiating component in HCC development [56]. EpCAM expresses in foetal livers, hepatic progenitor cells, carcinoma cells, etc., but not in mature hepatocytes [57]. In our study, EpCAM was selected by Cibersortx as a signature gene for cholangiocytes (Normal Liver atlas), proliferating cells (TME-Stroma), and bi-potent stem cells (TME-Immune), indicating the close relationship between these cell components and liver CSCs. Wnt–beta-catenin signalling activates EpCAM expression, which is associated with AFP elevation and foreshadows negative outcomes [58,59]. Pathway analysis by PRGOGENy suggests that hepatocytes and carcinoma cells are WNT active in the TME-Stroma atlas. PROGENy suggests high EGFR activities of bi-potent stem cells (TME-Immune) and proliferating cells (TME-Stroma). EGFR is responsible for the maintenance of multiple CSC phenotypes [60]. Survival analysis suggests the negative impacts of bi-potent stem cells on patient outcomes, providing alternative clinical evidence for the tumour-initiating hypothesis of EpCAM^+^ cells. Although the interpretation of these findings warrants careful consideration, our study demonstrates that deconvolution can also help understand the mechanism of cancer formation.

## 5. Conclusions

In this study, we deciphered the TME of liver cancer by estimating the cell fractions of a sample given a transcriptome. By estimating more than 20 cell types/subtypes within bulk RNA-seq data using three atlases and Cibersortx, we found disruptions of normal liver architecture, abnormal fibrogenesis and angiogenesis, as well as disturbed immune responses in HCC and CCA. Survival analysis demonstrated that five cell types/subtypes highly correlated with patient outcomes.

Deconvolution algorithm and emerging scRNA-seq atlases allow the decomposition of bulk RNA-seq data into cell-type fractions. By linking the cell fractions of samples and clinical follow-up information, we provide an innovative approach for the discovery of potential therapeutic targets. In the future, with the advent of more high-quality scRNA-seq atlases, deconvolution could be a powerful data mining tool for uncovering the intricate nature of the TME of liver cancer and revealing valuable information in the vast amount of available transcriptomic data.

## Figures and Tables

**Figure 1 cancers-15-00153-f001:**
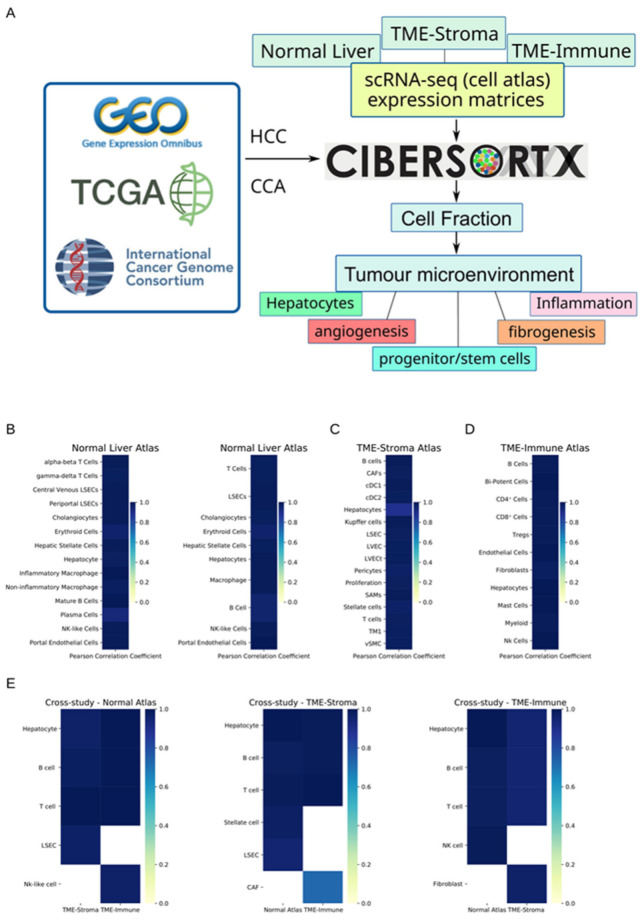
Study workflow and in silico validation of Cibersortx. (**A**) A general workflow of deciphering tumour microenvironment. Cibersortx estimates the cell fraction with scRNA-seq atlas and bulk RNA-seq/microarray data. Three expression matrices of scRNA-seq study were used as reference atlases. Through estimation of the fractions of more 20 cell types/subtypes, biological events could be inferred. (**B**–**D**) Performances of Cibersortx deconvolution in intra-study validation experiments. (**E**) Performances of Cibersortx deconvolution in cross-study validation experiments. The title of each panel indicates which reference atlas was used in the deconvolution, two ticks on the bottom indicate which dataset was used to generate the pseudo-bulk samples.

**Figure 2 cancers-15-00153-f002:**
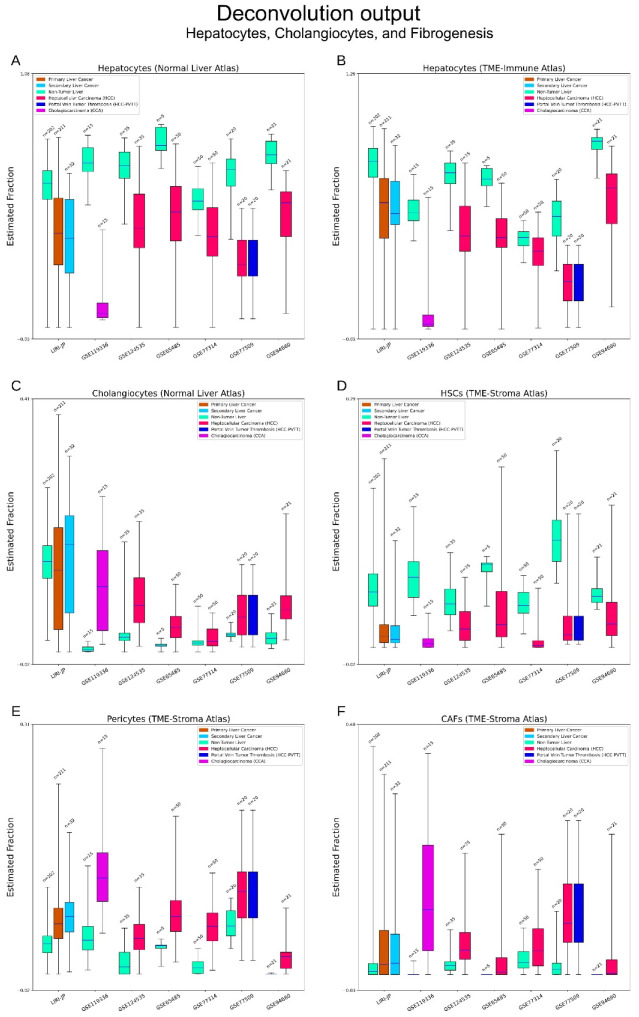
Deconvolution output—hepatocytes, cholangiocytes, and fibrogenesis. (**A**,**B**) A comparison of hepatocytes estimation by Cibersortx with two different atlases. Subtle differences of cell fractions can be seen between two experiments, but they arrived at the same conclusion, namely that hepatocytes decrease in cancerous tissue. In CCAs, this phenomenon is more prominent. (**C**) Estimation of cholangiocytes. Elevation of cholangiocytes can be widely seen in cancerous tissues. (**D**–**F**) Estimation of hepatic stellate cells (HSCs), pericytes and cancer-associated fibroblasts (CAFs). These three cell types are key components participating the fibrogenesis of liver cancer. HSCs are liver-specific pericytes, and often activated in liver cancer. Pericytes (broad sense) can be hardly detected in normal liver but are widely present in liver cancer. CAFs are uniquely characterised fibroblasts in cancer. They can hardly be detected in normal liver but are common in cancerous tissues.

**Figure 3 cancers-15-00153-f003:**
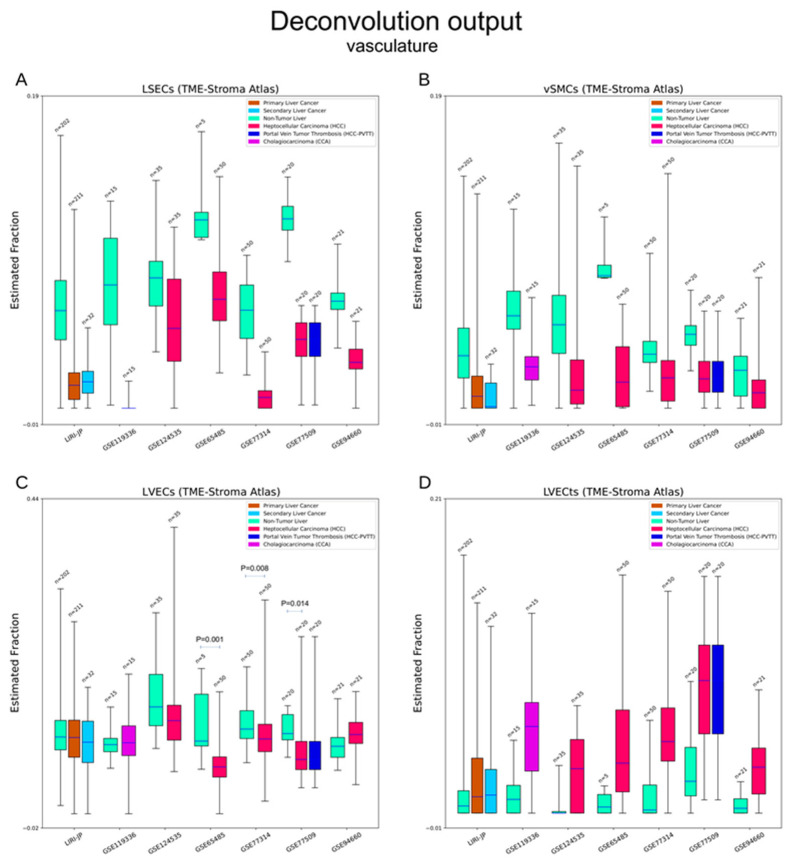
Deconvolution output—vasculature. (**A**–**D**) Estimation of liver sinusoidal endothelial cells (LSECs), vascular smooth muscle cells (vSMCs), liver vascular endothelial cells (LVECs), and tumour LVECs (LVECts). These four cell types/subtypes are key components of blood vessels in both normal and cancerous livers. Decreases of normal structural blocks (LSECs, vSMCs) and abnormal angiogenesis (LVECts) can be seen in liver cancers. Not all the differences are statistically significant. Pairs with *p* < 0.05 have been labelled.

**Figure 4 cancers-15-00153-f004:**
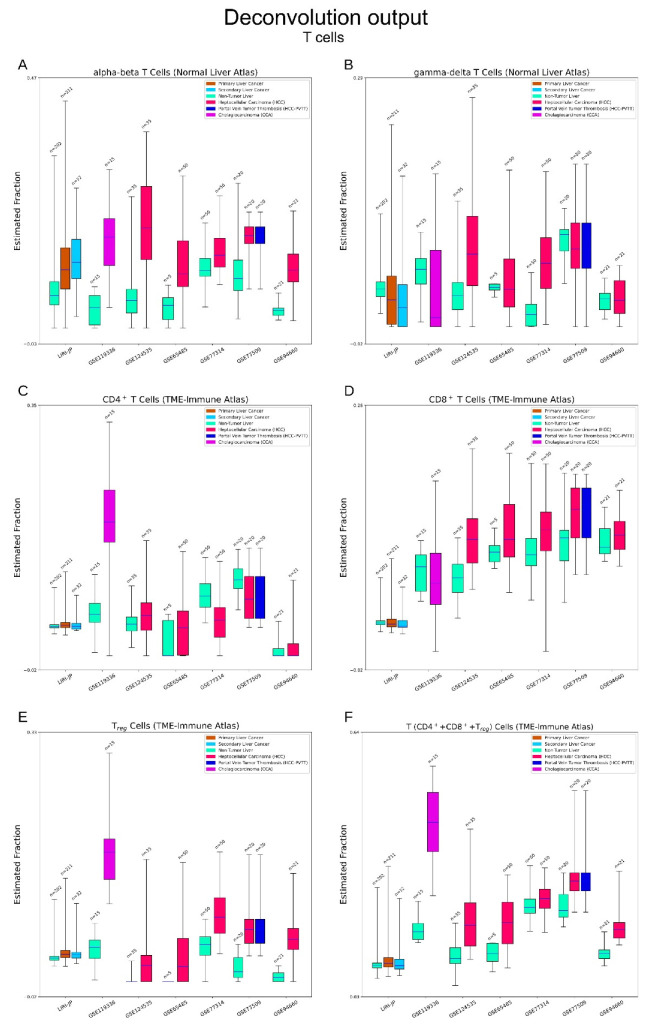
Deconvolution output—T cells. (**A**–**E**) Estimation of T cells. Alpha–beta (αβ) T cells and gamma–delta (γδ) T cells are from the Normal Liver atlas. CD4^+^/CD8^+^ T cells and regulatory T (T_reg_) cells are from the TME-Immune atlas. Overlap of cell type trees may exist between the two atlases. Obvious elevations of αβ T cells and obscure changes of γδ T cells can be seen in liver cancers. CD4^+^ T cells rise prominently in CCAs and CD8^+^ T cells moderately in HCCs. (**F**) Estimation of overall T cells. Higher activities of T cells can be seen in liver cancers.

**Figure 5 cancers-15-00153-f005:**
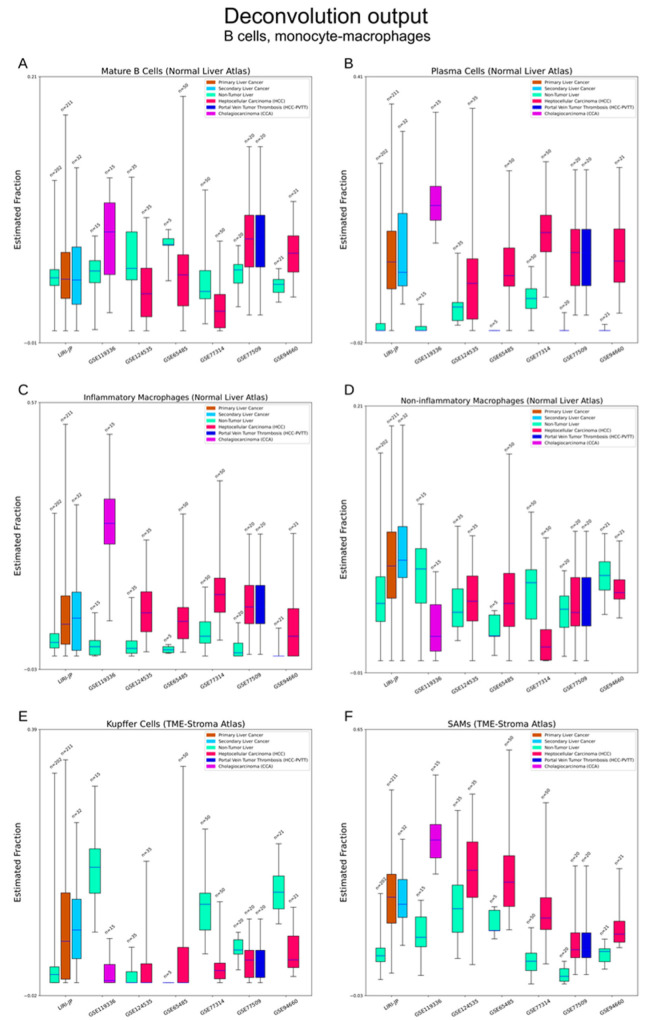
Deconvolution output—B cells, monocyte–macrophages. (**A**,**B**) Estimation of B cells. Plasma B cells can rarely be detected in normal livers but common in liver cancers. (**C**–**F**) Estimation of macrophages. The Normal Liver atlas separates macrophages into inflammatory and non-inflammatory subtypes. The TME-Stroma atlas identified Kupffer cells, scar-associated macrophages (SAMs) and tissue macrophages (TMs, Figure 6E). Overlap of cell type trees may exist between the two atlases. Increase of inflammatory macrophages can be seen in liver cancers. SAMs are a pathological variant of macrophages and participate in the process of liver fibrosis. Their elevation can be seen in liver cancers.

**Figure 6 cancers-15-00153-f006:**
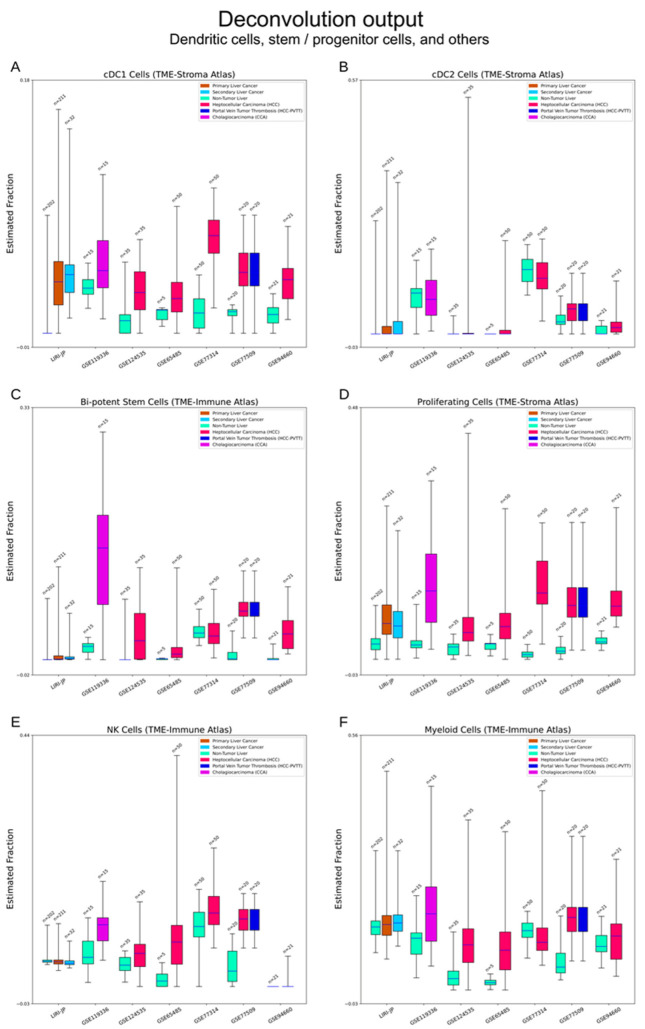
Deconvolution output—dendritic cells, stem/progenitor cells, and others. (**A**,**B**) Estimation of conventional dendritic cells (cDCs). Elevations of cDC1s can be seen in liver cancers. (**C**,**D**) Estimation of two types of proliferative cells. Bi-potent stem cells are a group of late-stage pluripotent cells with potential to differentiate into hepatocytes and cholangiocytes. The TME-Stroma atlas did not clarify the exact characteristics of proliferating cells. Its signature genes suggest its pluripotent origin. Elevations of these two cell types can be seen in liver cancers. (**E**,**F**) Estimation of natural killer (NK) cells and myeloid cells. NK cells belong to innate immune branch and myeloid cells are the hematopoiesis-originated immune branch. Deconvolution of these two cell types shows altered activities in liver cancers, but no direct association can be drawn.

**Figure 7 cancers-15-00153-f007:**
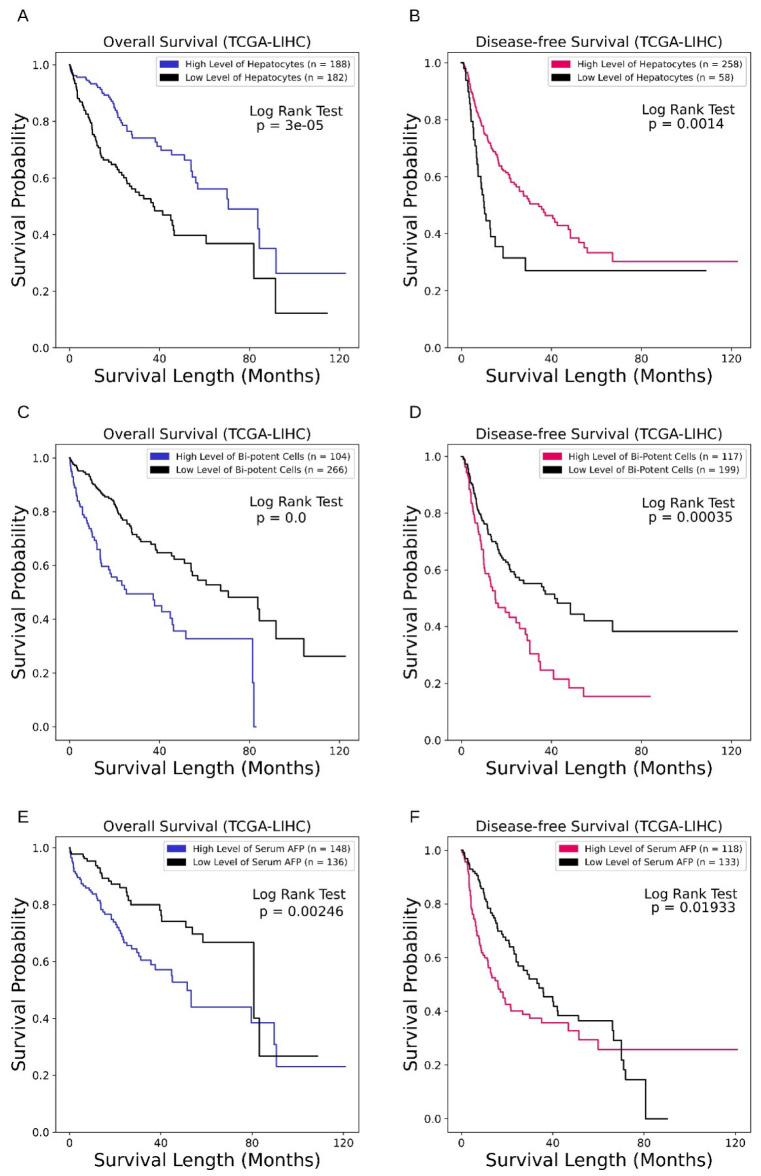
Impact of cell fractions on patients’ survivals. (**A**,**B**) Impact of hepatocyte fraction. High-level hepatocytes show both longer OS and DFS lengths. (**D**–**E**) Impact of bi-potent cell fraction. High-level bi-potent cells show lower OS and DFS lengths. (**C**,**F**) Impact of serum AFP at patient admission.

**Figure 8 cancers-15-00153-f008:**
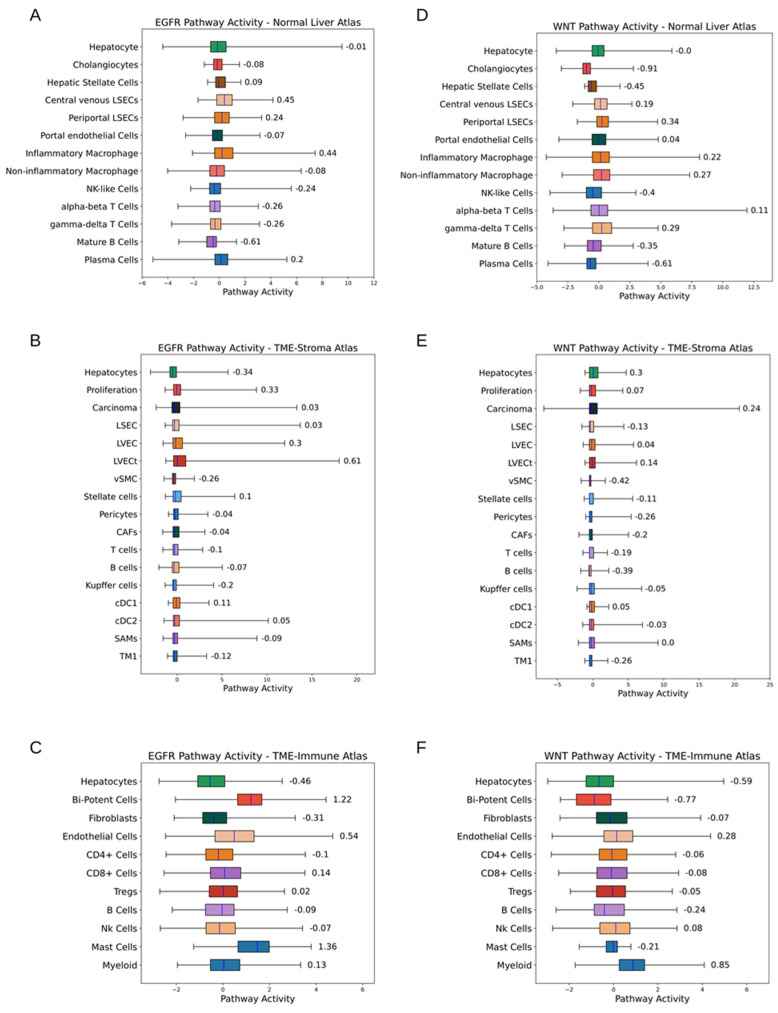
Pathway activities of cell types in the three atlases (estimated by PROGENy). (**A**–**C**) EGFR activities. Central venous LSECs, inflammatory macrophages, LVECt, bi-potent cells, mast cells, etc., are EGFR active, suggesting the involvement in angiogenesis, immune responses, and proliferation. (**D**–**F**) WNT activities. Periportal LSECs, inflammatory and non-inflammatory macrophages, gamma-delta T cells, hepatocytes (TME-Strom), carcinoma cells (TME-Stroma), myeloid cells, etc., are WNT active, suggesting the involvement in immune activities.

## Data Availability

All data is publicly available. The code used to reproduce this work is available from our GitHub (https://github.com/ErasmusMC-Bioinformatics/OIO_Shaoshi, accessed on 27 December 2022). The supplements have been uploaded to Zenodo (https://doi.org/10.5281/zenodo.7467268, accessed on 27 December 2022).

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
