# Peer review of "Deciphering Tumour Microenvironment of Liver Cancer through Deconvolution of Bulk RNA-Seq Data with Single-Cell Atlas"

_cancers, 2022, doi:10.3390/cancers15010153_

Round 1

Reviewer 1 Report

Zhang et al. introduced their novel approach to analyze the tumor microenvironment (TME) through deconvolution of bulk RNA-seq data with single-cell atlas. The paper is generally well written, but it seems too preliminary to be the reference in clinical practice. I understand the procedure elucidate the possible characteristics of TME, but I cannot find the actual significance of them in clinical situation. The authors should show the importance of this study, with comparing the results with other well-known biomarkers of HCC. I suppose this study might be better to be submitted to the journal of histopathology.

Author Response

Our study is invigorated by the emerging single-cell RNA-seq (scRNA-seq) atlases. ScRNA-seq atlases provide an unprecedented perspective on the heterogeneity of liver cancer. However, such analyses for all cancer patients are prohibitive. We aim to expand the discovery of scRNA-seq to the deposits of bulk RNA-seq studies. Our study provides a helicopter view of the microenvironment of liver cancer. Thus we think it fits the scope of the journal Cancers.

Reviewer 2 Report

In this manuscript, Zhang and colleagues analyzed major public datasets of transcriptome profiling which includes 8 RNA-Seq and 1 microarray datasets obtained in clinical studies of liver cancer. The goal of this analysis was to better understand tumor microenvironment underpins. The main result of this study is that liver cancer is characterized by a decrease of cell types that control normal liver functions and the elevation of cell types involved in fibrogenesis, abnormal angiogenesis and immune response. The authors also showed that the fractions of five cell types/subtypes significantly correlated with patient outcomes. These results provide a background for generation of potential therapeutic approaches.

In summary, this is a highly significant study which advances our knowledge of liver cancer and the role of tumor microenvironment in development of liver cancer. The work is well done, and the manuscript is well written.  I have only minor comments/suggestions.

Minor comments:

1)      It would be useful to discuss the results of this work in the light of reports showing that Cancer Associated Fibroblasts (CAFs) are a significant part of tumor microenvironment which is involved in development of lung metastases.  Although the authors mentioned the increase of CAFs, they may consider an expanded discussion of their finding focusing on development of metastases.

2)      The manuscript describes cell types that are involved in liver cancer.  It would strengthen the manuscript if the authors would look on main molecular pathways of liver cancer which correlate with cell types and clinical outcomes found in their studies. Particularly, discussion of beta-catenin and c-myc pathways might be included.

Author Response

Thanks for the suggestions.

Reply to #1: We separate the Discussion section according to the four topics, cancer stem cells, angiogenesis, fibrogenesis and immunity (Line 447-497). In each topic, we added more in-depth discussion about the critical components which influence carcinogenesis and receive interest for therapy development.

Reply to #2: Pathway analysis would enhance the implications of the findings in our study. We added pathway analysis by PROGENy in Figure 8 to discuss the relationship between cell types and EGFR/WNT signalling. PROGENy also predicts pathways such as MAPK, hypoxia, etc. We put them in supplements (S7 – Pathway Analysis) for the limited space of the main text.

Reviewer 3 Report

Summary:
This study applied a deconvolution method, Cibersortx, on liver cancer data, and found that five cell types are associated with patient outcomes.

Comments:

1. Since this paper only applied a method to data without any biological experiment / evidence to support the conclusion, I suggest applying 2 - 3 more supervised / unsupervised deconvolution methods to compare with Cibersortx, such as Tsoucas, Daphne, et al. "Accurate estimation of cell-type composition from gene expression data." Nature communications, Zaitsev, Konstantin, et al. "Complete deconvolution of cellular mixtures based on linearity of transcriptional signatures." Nature communications, Chen, Lulu, et al. "debCAM: a bioconductor R package for fully unsupervised deconvolution of complex tissues." Bioinformatics, or any other classic deconvolution methods. If some / all the methods give the similar result, then the conclusions would be more convincing.

2. The authors mentioned that they perform some validation experiments for the deconvolution with Cibersortx, and the results are in Supplementary information. However, I did not see there is any SI file. Also, the validation of Cibersortx is also important. so please consider to move some part of this validation from SI to main text.

3. Please improve the resolution of the legend and text in the figures, especially figure 2-6. 

Author Response

Thanks for the suggestions.

Reply to #1:

Cibersortx and MuSiC show state-of-the-art performances in various benchmarking studies (“Benchmarking of cell type deconvolution pipelines for transcriptomics data”. Nature Communications, Francisco, et.al; “A benchmark for RNA-seq deconvolution analysis under dynamic testing environments”, Genome Biology. Haijing Jin, et. al.). However, they are all deconvolution algorithms and share procedures (NNLS). To make the findings of Cibersortx more convincing, we also did a parallel series of analyses with support vector regression (SVR). Results of most cell types/subtypes by the two methods achieved the same conclusions (S1 – SVR Estimation.). We added notes about the interpretation of the results (main text, Line 355). Supporting Information describes the procedure of SVR with data and code online.

Reply to #2:

We added the link to supplements into the main text to avoid possible neglect (main text, Line 189, Line 532). Our study involves a lot of supporting materials. It seems hard to put the lengthy results of Cibersortx validation into the main text. For better readings, we combined supplementary figures and reduced the file numbers (S2 – In Silico Validation of Cibersortx). We also made one-to-one descriptions for supplements in Supporting Information.

Reply to #3:

The figures in the Word manuscript have been close to the upper limits of online pictures. This study originally generated figures in PDF format which can zoom in/out freely. We have packed these PDF copies into the supplements. Now they can be downloaded from the Zenodo.

Round 2

Reviewer 1 Report

The paper has been improved according to the reviewer's comments.

Reviewer 3 Report

I do not have further comments.